# Technical note: A new online tool for $\delta^{18}$O-temperature conversions

Daniel E. Gaskell[1], Pincelli M. Hull[1]

[1]Department of Earth & Planetary Sciences, Yale University, New Haven, CT 06511, USA

*Correspondence to*: Daniel E. Gaskell (daniel.gaskell@yale.edu)

**Abstract.** The stable oxygen isotopic composition of marine carbonates ($\delta^{18}O_c$) is one of the oldest and most widely-used paleothermometers. However, interpretation of these data is complicated by the necessity of knowing the $\delta^{18}O$ of the source seawater ($\delta^{18}O_w$) from which $CaCO_3$ is precipitated. The effect of local hydrography (the "salinity effect") is particularly difficult to correct for and may lead to errors of >10°C in sea-surface temperatures if neglected. A variety of methods for calculating $\delta^{18}O_w$ have been developed in the literature, but not all are readily accessible to workers. Likewise, temperature estimates are sensitive to a range of other calibration choices (such as calibration species and the inclusion or exclusion of carbonate ion effects) which can require significant effort to intercompare. We present an online tool for $\delta^{18}O$-temperature conversions which provides convenient access to a wide range of calibrations and methods from the literature. Our tool provides a convenient way for workers to examine the effects of alternate calibration and correction procedures on their $\delta^{18}O$-based temperature estimates.

## 1 Motivation

The stable oxygen isotopic composition of carbonates ($\delta^{18}O_c$) is one of the oldest and most widely-used paleothermometers and undergirds a wide variety of paleoceanographic research (for recent reviews, see Pearson, 2012; Sharp, 2017). Converting $\delta^{18}O_c$ to temperature is typically done using an empirical calibration in either a linear form such as

$$T = 16.5 - 4.80(\delta^{18}O_c - \delta^{18}O_w - 0.27), \tag{1}$$

(Bemis et al., 1998), or in a quadratic form such as

$$T = 16.0 - 5.17(\delta^{18}O_c - \delta^{18}O_w - 0.20) + 0.09(\delta^{18}O_c - \delta^{18}O_w - 0.20)^2, \tag{2}$$

(McCrea, 1950; as reformulated by Bemis et al., 1998), where $T$ is temperature (in °C), $\delta^{18}O_c$ is the oxygen isotope composition of the carbonate (as ‰ VPDB), and $\delta^{18}O_w$ is the oxygen isotope composition of the water in which the carbonate was precipitated (as ‰ VSMOW). Much of the complexity of using $\delta^{18}O$ as a paleothermometer arises from the need to know $\delta^{18}O_w$, which may vary both globally as a function of ice volume and locally at the sea surface as a function of regional hydrography (Rohling, 2013). Global variation can be estimated using independent records of sea level, so the global record of deep-water $\delta^{18}O$-based temperatures has been relatively well-established (Zachos et al., 2001; Cramer et al., 2009; Westerhold et al., 2020; Rohling et al., 2021; etc.) However, local variations in surface $\delta^{18}O_w$ are more difficult to predict,

rendering sea-surface temperature (SST) estimates from $\delta^{18}O$ less reliable than deep-water temperature estimates. To address this, a variety of methods have been developed in the literature to estimate surface $\delta^{18}O_w$.

Since modern surface $\delta^{18}O_w$ broadly covaries with latitude, a common approach has been to apply the modern latitudinal variation to a sample's paleolatitude (typically using the relationship fit from Southern Ocean data in Zachos et al., 1994 Eq. 1; or more recently the updated method of Hollis et al., 2019). However, this approach performs particularly poorly in the North Atlantic and other high northern latitudes, where local $\delta^{18}O_w$ can deviate significantly from the latitudinal mean (Fig. 1; Zachos et al., 1994; Gaskell et al., 2022; see also generally Tindall et al., 2010). It also assumes that the latitudinal gradient in $\delta^{18}O_w$ has not changed through time, which is contradicted by modeling. In warmer climates with an altered hydrological cycle, models predict that regional salinity contrasts should change due to alterations in the local ratio of evaporation to precipitation (Richter and Xie, 2010; Singh et al., 2016), with an analogous effect on $\delta^{18}O_w$ (Zhou et al., 2008; Tindall et al., 2010; Roberts et al., 2011; Zhu et al., 2020). In particularly extreme cases such as the Eocene, the theoretical difference between modern latitude-derived $\delta^{18}O_w$ (after Zachos et al., 1994 Eq. 1) and modeled local $\delta^{18}O_w$ at 6x preindustrial $p\mathrm{CO_2}$ (Zhu et al., 2020) yields a mean temperature error of 5 °C in the Southern Ocean (60–90 °S) or an astonishing mean temperature error of 41 °C above the Arctic Circle (66.5–90 °N; Figure 1).

An alternative approach is to obtain $\delta^{18}O_w$ more or less directly from isotope-enabled climate models (Zhou et al., 2008; Roberts et al., 2011; Gaskell et al., 2022). Several approaches have been adopted: drawing local $\delta^{18}O_w$ directly from model output (Roberts et al., 2011); using modeled zonal mean $\delta^{18}O_w$ for a particular paleolatitude (Zhou et al., 2008); using models as input to fit a generalized equation for predicting $\delta^{18}O_w$ from latitude and bottom-water temperature (Gaskell et al., 2022 Eq. S9); or, recently, a generalized method which uses bottom-water temperature to interpolate local $\delta^{18}O_w$ between models run at different $p\mathrm{CO_2}$ (Gaskell et al., 2022). While some authors have avoided these approaches altogether due to the uncertainty of modeled $\delta^{18}O_w$ (e.g., Hollis et al., 2012) or the possibility of introducing circularity into data-model comparisons (e.g., Hollis et al., 2019), model-derived $\delta^{18}O_w$ clearly captures information lost by simpler approaches and is therefore appropriate for some use-cases (Roberts et al., 2011).

Here, a new online tool for $\delta^{18}O$ temperature conversion is presented which automates a range of methods for $\delta^{18}O_w$ reconstruction and correction from the literature, improving the accessibility of advanced methods to workers generating $\delta^{18}O_c$ data.

## 2 Description

We present a new online tool for performing $\delta^{18}O_c$-temperature conversions which automates a range of methods from the literature. This tool is available at https://research.peabody.yale.edu/d18O/. The general workflow for using the tool is summarized in Figure 2; details on the methodology and reasoning behind each option are given below.

## 2.1 $\delta^{18}O_c$-temperature calibration

After manually entering or uploading a datasheet of $\delta^{18}O_c$ measurements in .csv format, users may select from one of 59 different calibrations from the literature (Bemis et al., 1998; Böhm et al., 2000; Bouvier-Soumagnac and Duplessy, 1985; Duplessy et al., 2002; Epstein et al., 1953; Erez and Luz, 1983; Farmer et al., 2007; Geffen, 2012; Godiksen et al., 2010; Grossman and Ku, 1986; Høie et al., 2004; Juillet-Leclerc and Schmidt, 2001; Kim and O'Neil, 1997; Kim et al., 2007; Lynch-Stieglitz et al., 1999; Malevich et al., 2019; Marchitto et al., 2014; McCrea, 1950; Mulitza et al., 2003; O'Neil et al., 1969;

Patterson et al., 1993; Reynaud-Vaganay et al., 1999; Rosenheim et al., 2009; Shackleton, 1974; Storm-Suke et al., 2007; Thorrold et al., 1997; Tremaine et al., 2011; White et al., 1999; Willmes et al., 2019). All data are expressed with $\delta^{18}O_c$ in units of ‰ VPDB and $\delta^{18}O_w$ in units of ‰ VSMOW, with any standard interconversions expected by the chosen calibration performed automatically. Standard interconversion is notably inconsistent in the literature, with many paleoceanographic papers employing the relationship $\partial^{18}O_{VPDB} = \partial^{18}O_{VSMOW} - 0.27‰$ (Hut, 1987) while many geochemical papers employ the

incompatible relationship $\partial^{18}O_{VPDB} = 0.97001\,\partial^{18}O_{VSMOW} - 29.99‰$ (Brand et al., 2014). The former is actually the isotopic offset between the related VPDB-$CO_2$ and VSMOW-$CO_2$ scales, but the difference is unimportant so long as all data are treated in the manner the calibration expects, as our tool ensures. A full list of included calibrations and standard conversions are given in Tables 1–3.

Where applicable, we use the standardized reformulations of Bemis et al. (1998) and Willmes et al. (2019), or exact algebraic

rearrangements of the original equations. For the bayfox core-top calibrations of Malevich et al. (2019), the standard bayfox tool re-fits the calibration coefficients with every run. Since this is computationally expensive, we instead use the linear calibration coefficients fit by runs of bayfoxr 0.0.1 directly in linear functions of the form of Eq. 1 (see Table 1). These yield results equivalent to the full fitting process within numerical error (mean residual = ±0.02 °C, identical to the mean scatter between replicates of the full bayfox fit).

## 2.2 Global $\delta^{18}O_w$ estimation

Users may specify global $\delta^{18}O_w$ manually or choose to draw $\delta^{18}O_w$ by sample age from of 12 different timeseries of global $\delta^{18}O_w$ from the literature (from Cramer et al., 2011; Henkes et al., 2018; Meckler et al., 2022; Miller et al., 2020; Modestou et al., 2020; Rohling et al., 2021; Veizer and Prokoph, 2015). These records are typically constructed by assuming that the benthic $\delta^{18}O$ record reflects a combination of temperature and ice volume and then subtracting out an independent record of

temperature (e.g., using Mg/Ca-based bottom-water temperatures; Cramer et al., 2011) or ice volume (e.g., using a multi-proxy sea level reconstruction; Rohling et al., 2021) to determine the residual $\delta^{18}O_w$. Which global $\delta^{18}O_w$ record is most realistic remains a contentious topic in the literature, with sea-level and Mg/Ca-based records (e.g., Cramer et al., 2011; Rohling et al., 2021) predicting up to ~1‰ lower $\delta^{18}O_w$ for much of the Cenozoic than records based on clumped isotope paleothermometry (Meckler et al., 2022; see also Agterhuis et al., 2022). We provide both classes of record here for comparison by the user.

Records are mapped to the user data's ages by linear interpolation. The Δ47-based $\delta^{18}O_w$ records of Meckler et al. (2022) included in our tool were generated by interpolating the authors' original results to 0.1 Ma resolution using the Monte Carlo LOESS method and parameters described in the original publication (Meckler et al., 2022).

All built-in $\delta^{18}O_w$ and temperature records are internally converted to four different timescales, so the user can select the timescale consistent with their data: GTS2004 (Gradstein et al., 2005), GTS2012 (Gradstein et al., 2012), GTS2016 (Ogg et

al., 2016), and GTS2020 (Gradstein et al., 2020). These timescale conversions are performed by linear interpolation between magnetochron boundaries; dataset files can be found on the project GitHub.

## 2.3 Local $\delta^{18}O_w$ estimation

The user may select a method for estimating local $\delta^{18}O_w$. These are as follows: performing no local correction; using modern $\delta^{18}O_w$ from each sample's location and a specified depth (after LeGrande and Schmidt, 2006); using reconstructed Late

Holocene or Last Glacial Maximum surface $\delta^{18}O_w$ from each sample's location (model output from Tierney et al., 2020); using $\delta^{18}O_w$ estimated from latitude alone (after Zachos et al., 1994 Eq. 1; or the method of Hollis et al., 2019); using $\delta^{18}O_w$ estimated from latitude and bottom-water temperature (after Gaskell et al., 2022 Eq. S9); or using $\delta^{18}O_w$ estimated from isotope-enabled climate models (GCMs, after the method of Gaskell et al., 2022, presently provided using the datasets of Miocene and Eocene paleogeography used in that publication).

For methods which draw from an existing dataset of $\delta^{18}O_w$, the user may specify a number of degrees latitude/longitude or great-circle radius to average over in order to capture a regional mean when the exact paleocoordinates or local hydrography may not be known. To help determine site locations at the time of deposition, an option is also provided to automatically perform paleocoordinate rotations using the GPlates Web Service (Müller et al., 2018). Ages passed to GPlates are rounded to the nearest 100 ka to reduce the number of API calls.

Our tool does not currently implement any automated consideration of seasonal variation in local $\delta^{18}O_w$, as this is generally treated as negligible by standard methodologies or implicitly baked into the calibration by calibrating against mean annual temperatures and $\delta^{18}O_w$ values (e.g., Malevich et al., 2019).

## 2.4 Carbonate chemistry effects

Because $\delta^{18}O_c$ is known to vary with aqueous carbonate chemistry (the "carbonate ion effect"; Spero et al., 1997; Bijma et al.,

1999; Ziveri et al., 2012), users may also specify a carbonate ion correction factor. This is performed by adjusting $\delta^{18}O_c$ with the linear relationship

$$\delta^{18}O_c' = \delta^{18}O_c - (s[CO_3^{2-}] - 200s), \qquad (3)$$

where $\delta^{18}O_c$ is the uncorrected oxygen isotope composition of the carbonate, $\delta^{18}O_c'$ is the corrected oxygen isotope composition of the carbonate, $s$ is the selected slope of the effect (in ‰ VPDB per µmol L$^{-1}$ CO$_3^{2-}$), and [CO$_3^{2-}$] is the concentration of

carbonate ion in solution (in µmol kg$^{-1}$). This relationship yields no correction when [CO$_3^{2-}$] = 200 µmol kg$^{-1}$, an approximation

of the mean modern surface value (after the long-term record of Zeebe and Tyrrell, 2019). The user may specify $[CO_3^{2-}]$ manually or select a published long-term record of $[CO_3^{2-}]$ (Tyrrell and Zeebe, 2004; Zeebe and Tyrrell, 2019).

## 2.5 Tool output

On completion, the tool presents a formatted table of the resulting temperatures, along with any intermediate values (such as estimated $\delta^{18}O_w$) which were required to generate them. Any rows with potential errors (e.g., paleocoordinates which do not yield a valid $\delta^{18}O_w$ estimate or temperatures which exceed the data range of the calibration) are highlighted in color and flagged with warning text, which appears in an adjacent column. For reference, a short summary of methods is also generated, including relevant equations and a complete bibliography of citations in both text and BibTeX formats for the methods employed in each run.

It should be noted that, while the tool automates the process of applying a given calibration method, the user is still responsible for pre-screening their data for diagenetic alteration or other external biases. For example, use of $\delta^{18}O$ data from foraminifera must consider factors such as diagenetic recrystallization, depth habitat, shell size, and the presence of gametogenic calcite (for a review, see Pearson, 2012).

## 3 Concluding remarks

Our tool provides a convenient way for workers to perform $\delta^{18}O$-temperature conversions and explore the sensitivity of their results to different calibrations, corrections, and $\delta^{18}O_w$-reconstruction methods by successively trying different options in the interface. By allowing data-generators to rapidly generate multiple temperature estimates for their records with different underlying assumptions, our tool allows workers to quickly understand and quantify the effects of different assumptions on the resulting temperature estimates.

## Code availability

An online version of the most current release of our tool is maintained at https://research.peabody.yale.edu/d18O/. Source code (Javascript and PHP) is available from the project's GitHub repository at https://github.com/danielgaskell/d18Oconverter (release DOI: 10.5281/zenodo.7946599).

## Author contribution

DG and PH conceptualized the tool. DG wrote the software. DG and PH contributed to the manuscript writing.

## Competing interests

The authors declare that they have no conflict of interest.

## Acknowledgements

PH acknowledges funding from NSF Award #1702851. We thank Matthew Huber, Charlotte L. O'Brien, Gordon N. Inglis, R. Paul Acosta, and Christopher J. Poulsen for discussions which contributed to the design of this tool and associated methodologies. We thank Brett Metcalfe and an anonymous referee for their constructive comments which improved the manuscript. We acknowledge institutional support from the Yale Peabody Museum and assistance from Nelson Rios in hosting the tool.

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

**Figure 1: Effect of estimating SST using measured/modelled local $\delta^{18}O_w$ rather than the latitude-based approximation of Zachos et al. (1994) Eq. 1. Modern: comparison with mean annual $\delta^{18}O_w$ <50 m depth (after LeGrande and Schmidt, 2006). Last Glacial Maximum (LGM): comparison with inferred annual surface $\delta^{18}O_w$ at the LGM (Tierney et al., 2020). Miocene: comparison with CESMv1.2_CAM5 model run at 400 ppm $CO_2$ with Miocene paleogeography (Gaskell et al., 2022). Eocene: comparison with CESM_1.2_CAM5 model run at 6x preindustrial $CO_2$ with Eocene paleogeography (Zhu et al., 2020). Temperatures are calculated assuming a slope of 4.80 °C ‰$^{-1}$ (Bemis et al., 1998).**

**Figure 2: General workflow for using the tool. (Each box may reflect multiple sub-options.)**

**Table 1:** Linear and quadratic $\delta^{18}O$:temperature calibrations of the form $T = a + b\,\Delta^{18}O_c + c\left(\Delta^{18}O_c\right)^2$, where $\Delta^{18}O_c = \partial^{18}O_c - \partial^{18}O_w$ with the given VSMOW conversion factor first added to $\partial^{18}O_w$ to convert VSMOW into the format expected by the calibration.

| Reference | Material | Method | a | b | b | VSMOW to VPDB | Bounds (°C) | Yield |
|---|---|---|---|---|---|---|---|---|
| Bemis et al. (1998) | *Orbulina universa* high light (HL) | Culture | 14.9 | −4.80 | — | −0.27‰ | 15–25 | *In situ* temperature |
| " | *Orbulina universa* low light (LL) | " | 16.5 | −4.80 | — | −0.27‰ | " | *In situ* temperature |
| " | *Orbulina universa* HL + LL mean | " | 15.7 | −4.80 | — | −0.27‰[§] | " | *In situ* temperature for generic photosymbiotic planktonic foraminifera |
| " | *Globigerina bulloides* (11th chamber) | " | 12.6 | −5.07 | — | −0.27‰ | 15–24 | *In situ* temperature |
| " | *Globigerina bulloides* (12th chamber) | " | 13.2 | −4.89 | — | −0.27‰ | " | *In situ* temperature |
| " | *Globigerina bulloides* (13th chamber) | " | 13.6 | −4.77 | — | −0.27‰ | " | *In situ* temperature |
| Bouvier-Soumagnac and Duplessy (1985) | *Orbulina universa* | Culture | 16.4 | −4.67 | — | −0.20‰* | 20–24.6 | *In situ* temperature |
| " | *Orbulina universa* | Plankton tow regression | 15.4 | −4.81 | — | −0.20‰* | 20–29.5 | " |
| " | *Globorotalia menardii* | Plankton tow regression | 14.6 | −5.03 | — | −0.20‰[§] | 22.6–29.2 | " |
| " | *Neogloboquadrina dutertrei* | Plankton tow regression | 10.5 | −6.58 | — | −0.20‰[§] | 24.6–30.6 | " |
| Duplessy et al. (2002) | *Cibicides* spp. | Core-top regression | 12.75 | −3.60 | — | −0.20‰[†] | −2–13 | Bottom-water temperature |
| Epstein et al. (1953) | Mixed biogenic carbonates | Field sample regression | 16.5 | −4.30 | 0.14 | −0.27‰[ǀ] | 7–30 | *In situ* temperature |
| Erez and Luz (1983) | *Trilobatus sacculifer* | Culture | 17.0 | −4.52 | 0.03 | −0.22‰* | 14–30 | *In situ* temperature |
| Farmer et al. (2007) | *Globigerinoides ruber* (white) | Core-top regression | 15.4 | −4.78 | — | −0.27‰ | (modern ocean) | Mean annual *in situ* temperature |
| " | *Globigerinoides ruber* (pink) | " | 14.7 | −4.86 | — | " | " | " |
| " | *Trilobatus sacculifer* | " | 16.2 | −4.94 | — | " | " | " |
| " | *Orbulina universa* | " | 16.5 | −5.11 | — | " | " | " |
| " | *Pulleniatina obliquiloculata* | " | 16.8 | −5.22 | — | " | " | " |

| Reference | Material | Method | | | | | | |
|---|---|---|---|---|---|---|---|---|
| " | *Globorotalia menardii* | " | 16.6 | –5.20 | — | " | " | " |
| " | *Neogloboquadrina dutertrei* | " | 14.6 | –5.09 | — | " | " | " |
| " | *Neogloboquadrina tumida* | " | 13.1 | –4.96 | — | " | " | " |
| Grossman and Ku (1986) | Mixed molluscs | Core-top regression | 21.8 | –4.69 | — | 0.20‰ | 6–22 | Bottom-water temperature |
| " | *Hoeglundina elegans* | " | 20.6 | –4.38 | — | " | 2.5–20 | " |
| Juillet-Leclerc and Schmidt (2001) | *Porites* spp. | Field sample regression | 9.25 | –4.00 | — | −0.27‰[§] | 20–30 | Annual *in situ* temperature |
| Kim and O'Neil (1997) | Inorganic calcite | Precipitation | 16.1 | –4.64 | 0.09 | −0.27‰[*] | 0–40 | *In situ* temperature |
| Lynch-Stieglitz et al. (1999) | *Cibicidoides + Planulina* | Core-top regression | 16.1 | –4.76 | — | −0.27‰[§] | 4.1–25.6 | Bottom-water temperature |
| Malevich et al. (2019) | Mixed planktonic foraminifera | Core-top regression | 11.8790 | –4.0562 | — | 0‰[‡] | 0–29.5 | Mean annual sea-surface temperatures (SSTs) |
| " | *Globigerinoides ruber* | " | 13.0681 | –5.2605 | — | " | " | " |
| " | *Trilobatus sacculifer* | " | 12.4053 | –6.3458 | — | " | " | " |
| " | *Globigerina bulloides* | " | 16.6159 | –4.1291 | — | " | " | " |
| " | *Trilobatus sacculifer* | " | 12.4053 | –6.3458 | — | " | " | " |
| " | *Neogloboquadrina incompta* | " | 17.9531 | –5.7401 | — | " | " | " |
| " | *Neogloboquadrina incompta* | " | 19.8109 | –4.9853 | — | " | " | " |
| McCrea (1950) | Inorganic calcite | Precipitation | 16.0 | –5.17 | 0.09 | −0.20‰[*] | 14–57 | *In situ* temperature |
| Mulitza et al. (2003) | Mixed planktonic foraminifera | Plankton tow regression | 14.32 | –4.28 | 0.07 | −0.27‰ | –2–31 | *In situ* temperature |
| " | *Trilobatus sacculifer* | " | 14.91 | –4.35 | — | " | 16–31 | " |
| " | *Globigerinoides ruber* (white) | " | 14.20 | –4.44 | — | " | 16–31 | " |
| " | *Globigerina bulloides* | " | 14.62 | –4.70 | — | " | 1–25 | " |
| " | *Neogloboquadrina pachyderma* | " | 12.69 | –3.55 | — | " | 1–25 | " |
| O'Neil et al. (1969) | Inorganic calcite | Precipitation | 16.9 | –4.38 | 0.10 | −0.20‰[*] | 0–500 | *In situ* temperature |

| Reynaud-Vaganay et al. (1999) | *Stylophora pistillata* | Culture | 16.15 | −7.69 | — | 1.29‰[§] | 21–29 | *In situ* temperature |
| " | *Acropora* spp. | " | 19.81 | −3.70 | — | " | " | " |
| Rosenheim et al. (2009) | *Ceratoporella nicholsoni* | Culture | 16.1 | −6.5 | — | 0‰ | 23–27.5 | *In situ* temperature |
| Shackleton (1974) | *Uvigerina* spp. | Core-top regression | 16.9 | −4.0 | — | −0.20‰* | 0.8–7 | Bottom-water temperature |

*Reformulated by Bemis et al. (1998)

[†]Reformulated by Mulitza et al. (2003)

[‡]Reformulated in this work by extracting the linear coefficients from the Bayesian posterior values

[§]Rearranged by this work

[ǀ]Reformulated by Bemis et al. (1998), with VSMOW correction after Grossman (2012)

**Table 2: Logarithmic δ18O:temperature calibrations of the form $1000 \ln \alpha = a\left(10^3\,\mathrm{TK}^{-1}\right) - b$, where TK is temperature (in Kelvin) and $\alpha$ is the fractionation factor $\alpha = \frac{\delta^{18}O_c + 1000}{\delta^{18}O_w + 1000}$. The temperature solution for this form (in °C) is**

$$T = \frac{a \times 10^3}{1000 \ln\left(\frac{\partial^{18}O_c + 1000}{\partial^{18}O_w + 1000}\right) + b} - 273.15$$

**where the relationship $\partial^{18}O_{\mathrm{VPDB}} = 0.97001\,\partial^{18}O_{\mathrm{VSMOW}} - 29.99$ (Brand et al., 2014) is first applied to either convert δ18O$_c$ from VPDB to VSMOW, or to convert δ18O$_w$ from VSMOW to VPDB, as required by the calibration. (The value requiring conversion is indicated in the "Convert which" column.)**

| Reference | Material | Method | a | b | Convert which | Bounds (°C) | Yield |
|---|---|---|---|---|---|---|---|
| Böhm et al. (2000) | *Ceratoporella nicholsoni* | Field sample regression | 18.45 | 32.54 | δ18O$_w$ | 3–28 | *In situ* temperature |
| Geffen (2012) | *Pleuronectes platessa* otoliths | Culture | 15.99 | 24.25 | δ18O$_w$ | 11–17 | *In situ* temperature |
| Godiksen et al. (2010) | *Salvelinus alpinus* otoliths | Culture | 20.43 | 41.14 | δ18O$_w$ | 2–14* | *In situ* temperature |
| Høie et al. (2004) | *Gadus morhua* otoliths | Culture | 16.75 | 27.09 | δ18O$_w$ | 6–20 | *In situ* temperature |
| Kim et al. (2007) | Inorganic aragonite | Precipitation | 17.88 | 31.14 | δ18O$_c$ | 0–40 | *In situ* temperature |
| Patterson et al. (1993) | Mixed freshwater lake fish | Field sample regression | 18.56 | 33.49 | δ18O$_w$ | 3.2–30.3 | *In situ* temperature |
| Storm-Suke et al. (2007) | *Salvelinus* spp. otoliths | Field sample regression | 20.69 | 41.69 | δ18O$_w$ | 2.3–11.8 | *In situ* temperature |
| Thorrold et al. (1997) | *Micropogonias undulatus* otoliths | Culture | 18.57 | 32.54 | δ18O$_w$ | 18.2–25* | *In situ* temperature |
| Tremaine et al. (2011) | Speleothem calcite | Field sample regression | 16.1 | 24.6 | δ18O$_w$ | 16–21.5 | *In situ* temperature |

| White et al. (1999) | *Lymnaea peregra* | Culture | 16.74 | 26.39 | $\delta^{18}O_w$ | 8–24 | *In situ* temperature |
| Willmes et al. (2019) | *Hypomesus transpacificus* otoliths | Culture | 18.39 | 34.56 | $\delta^{18}O_w$ | 16.4–20.5 | *In situ* temperature |

*Reformulated by Willmes et al. (2019)

**Table 3: $\delta^{18}O$:temperature calibrations in other forms. $\Delta^{18}O_c = \partial^{18}O_c - \partial^{18}O_w$; the VSMOW/VPDB conversion is included in the equations below, with no further conversion required.**

| Reference | Material | Method | Equation | Bounds (°C) | Yield |
|---|---|---|---|---|---|
| Marchitto et al. (2014) | *Cibicidoides + Planulina* | Core-top regression | $T = \dfrac{0.245 - \sqrt{0.045461 + 0.0044\,\Delta^{18}O_c}}{0.0022}$ | –0.6–25.6 | Bottom-water temperature |
| " | *Uvigerina peregrina* | " | $T = \dfrac{0.242 - \sqrt{0.046468 + 0.0032\,\Delta^{18}O_c}}{0.0016}$ | –1.5–16.9* | " |
| " | *Uvigerina* spp. | " | Recommended method: subtract 0.47‰ from $\delta^{18}O_c$ and use *Cibicidoides* eq. above | n/a | " |
| " | *Hoeglundina elegans* | " | $T = \dfrac{0.242 - \sqrt{0.053176 + 0.0012\,\Delta^{18}O_c}}{0.0006}$ | 2.6–25.6* | " |

*Rearranged by this work