# Peer review of "Technical note: A new online tool for $\delta^{18}$ O-temperature conversions"

_Climate of the Past, 2022_

## Referee Comment (RC1)

**Review figures:**

*Table 1 - 2*  *Palaeotemperature equations*

| Reference (Original) | Reference (Rearrranged or reformated) | Material | Calibration Temperature (°C) Minimum | Maximum | a | b | (c - sw) | c | (c - sw)$^2$ | VSMOW to VPDB (‰) | Comments |
|---|---|---|---|---|---|---|---|---|---|---|---|
| McCrea, 1950 | | Synthetic calcite | | | 16 | 5.17 | o | 0.09 | o | -0.2 | |
| Epstein et al., 1953 | (Epstein & Mayeda, 1953) | Mollusk Shell | 7.2 | 29.5 | 16.50 | 4.30 | o | 0.14 | o | -0.27 | Conversion is -0.27‰ as it was directly standardized with PDB-derived $CO_2$ |
| - | Craig, 1965 | Mollusk Shell | | | 16.90 | 4.20 | o | 0.13 | o | -0.20 | |
| - | Shackleton and Opdyke, 1973 | Mollusk Shell | 7.2 | 29.5 | 16.90 | 4.38 | o | 0.13 | o | | Minor variant published in Malaizé and Caley, 2009 |
| - | Anderson and Arthur, 1983 | Mollusk Shell | 7.2 | 29.5 | 16.00 | 4.14 | o | 0.13 | o | | Revision with $\delta^{18}O$w referenced to VSMOW |
| O'Neil et al., 1969 | Shackleton, 1974 | Synthetic calcite | 0 | 500 | 16.90 | 4.38 | o | 0.10 | o | -0.20 | Quadratic approximation from the original 1000lnα notation, calibrated with *Uvigerina* |
| - | Hays and Grossman, 1991 | Synthetic calcite | 0 | 60 | 15.70 | 4.36 | o | 0.12 | o | | Minor variant with correction of Friedman and O'Neil, 1977 (Grossman, 2012) |
| Horibe and Oba, 1972 | | Cultured mollusc *Patinopecten yessoensis* (Mutsu Bay, Japan) | 4.5 | 23.3 | 17.04 | 4.34 | o | 0.16 | o | -0.20 | |
| Erez and Luz, 1983 | | Cultured *Globigerinoides sacculifer* (50-90% growth under lab. Conditions) | 14 | 30 | 16.998 | 4.52 | o | 0.028 | o | | Minor variant in literature |
| - | Pearson, 2012 | Cultured *Globigerinoides sacculifer* (50-90% growth under lab. Conditions) | 14 | 30 | 17.00 | 4.52 | o | 0.03 | o | -0.22 | Overestimation of temperatures by 2°C |
| Bouvier-Soumagnac and Duplessy, 1985 | | *Orbulina universa* cultured | | | 16.40 | 4.67 | o | - | x | -0.20 | |
| Bouvier-Soumagnac and Duplessy, 1985 | | *Orbulina universa* (Indian Ocean) | | | 15.40 | 4.81 | o | - | x | -0.20 | |
| Grossman and Ku, 1986 | | Biogenic aragonite | 2.6 | 22 | 20.60 | 4.34 | o | - | x | 0.20 | |
| - | Hudson and Anderson, 1989 | Biogenic aragonite | 2.6 | 22 | 19.70 | 4.34 | o | - | x | - | water values cast in terms of VSMOW (Grossman, 2012) |
| Kim and O'Neil, 1997 | Bemis et al., 1998 | Synthetic calcite | 10 | 40 | 16.10 | 4.64 | o | 0.09 | o | -0.27 | Quadratic approximation using a least square regression from the original 1000lnα notation, offset of ~ - 2°C for photosymbiotic species (if analogous with modern calcification in low-pH microenvironment |
| - | Peeters et al., 2002 | Synthetic calcite | 10 | 40 | 15.20 | 4.60 | o | 0.09 | o | -0.27 | Quadratic approximation from the original 1000lnα notation, offset of ~ - 2°C for photosymbiotic species (if analogous with modern calcification in low-pH microenvironment |
| - | Grossman 2012 | Synthetic calcite | 10 | 40 | 13.70 | 4.54 | o | 0.09 | o | | Quadratic approximation from the original 1000lnα notation, using 1000lnα = 10.44 |
| Bemis et al., 1998 | | Cultured *Orbulina universa* | 15 | 25 | 14.90 | 4.80 | o | - | x | -0.27 | High light (HL) = (>380 µEinst m$^{-2}$ s$^{-1}$) |
| Bemis et al., 1998 | | Cultured *Orbulina universa* | 15 | 25 | 16.50 | 4.80 | o | - | x | -0.27 | Low Light (LL) = (20-30 µEinst m$^{-2}$ s$^{-1}$) |
| Bemis et al., 1998 | | Cultured *Globigerina bulloides* (11-chambered shell) | 14.5 | 24 | 12.60 | 5.07 | o | - | x | -0.27 | Mass balance relationship, where δ18O values of the first 10 chambers are estimated at the experimental temperature via interpolation of 10-chambered shells collected at 16°C (Spero and Lea, 1996) and 22°C (Bemis et al., 1998) |
| Bemis et al., 1998 | | Cultured *Globigerina bulloides* (12-chambered shell) | 14.5 | 24 | 13.20 | 4.89 | o | - | x | -0.27 | Mass balance relationship, where δ18O values of the first 10 chambers are estimated at the experimental temperature via interpolation of 10-chambered shells collected at 16°C (Spero and Lea, 1996) and 22°C (Bemis et al., 1998) |
| Bemis et al., 1998 | | Cultured *Globigerina bulloides* (13-chambered shell) | 14.5 | 24 | 13.60 | 4.77 | o | - | x | -0.27 | Mass balance relationship, where δ18O values of the first 10 chambers are estimated at the experimental temperature via interpolation of 10-chambered shells collected at 16°C (Spero and Lea, 1996) and 22°C (Bemis et al., 1998) |
| Lynch-Stieglitz et al., 1999 | | In-situ *Cibicidoides* and *Planulina* (Surface sediments, Little Bahama Bank) | 4.1 | 25.6 | 16.10 | 4.76 | o | - | x | -0.27 | |
| - | Cramer et al., 2011 | In-situ *Cibicidoides* and *Planulina* (Surface sediments, Little Bahama Bank) | 4 | 26 | 16.10 | 4.76 | o | - | x | -0.27 | |
| Mielke, 2001 | Spero et al., 2003 | Cultured *Globorotalia menardii* | ? | ? | 14.90 | 5.13 | o | - | x | -0.27 | |
| Spero et al., unpublished | Spero et al., 2003 | Cultured *Globigerinoides sacculifer* | ? | ? | 12.00 | 5.57 | o | - | x | -0.27 | High light (HL) |
| Mulitza et al., 2003 | | In-situ *Globigerinoides sacculifer* | 16 | 31 | 14.91 | 4.35 | o | - | x | -0.27 | High light (HL) |

**Figure 1:** Example of a table format for summarizing palaeotemperature calibrations including the calibration temperature minimum and maximum, material and format of the equation. The o and x in (c-sw) and (c-sw)^2 columns denote either their presence or absence from the formula. From Metcalfe (2013).

**Review References**

Metcalfe, B. (2013, December 17). Planktonic foraminifera: From production to preservation of the oceanographic signal. Amsterdam, Netherlands: Vrije Universiteit Amsterdam. https://research.vu.nl/en/publications/planktonic-foraminifera-from-production-to-preservation-of-the-oc

---

## Author Response (AR1)

**SPECIFIC COMMENTS TO EDITOR**

We thank the editor and both reviewers for their positive, constructive, and thorough comments on our preprint manuscript. We have adopted the majority of these requests as-is, with specific comments below; the most significant changes in this version of the manuscript and tool are as follows:

1. In response to comments by Reviewer 1, we have added to the manuscript a variety of methodological details which were previously relegated to the code or associated documentation (e.g., exactly how we implement the Malevich et al. 2019 Bayesian calibration, or how interpolation between points is handled). All papers used by the tool are now cited in-text, not only in the methodology output of the tool itself. We have also added a table of commonly-used calibrations for reference, as suggested by Reviewer 1.
2. We have made a number of feature updates to the tool itself, including: 1) BibTeX citation output; 2) more detailed methodology output; 3) versioning; 4) a new d18Osw dataset, Meckler et al. (2022); and various other small suggested tweaks.
3. In response to comments from both reviewers, we have omitted the majority of the former content of Section 3, which applied the tool to the DeepMIP proxy database to demonstrate the relative effects of different methodologies on estimated temperatures. We agree with the reviewers that this section was too disconnected from the main text, which is now more tightly focused on the details of the tool itself, although we retain a brief discussion of the significance of spatially-explicit d18Osw estimation methods as part of the introduction.

**REVIEWER COMMENT #1**

In "Technical note: A new online tool for d18O" Gaskell and Hull (2022) report on an online tool they have produced for helping practitioners convert d18Oc into palaeotemperature. The tool allows user to define multiple options including their favorite d18Oc-temperature calibration, site latitude /longitude, and estimated d18Osw to produce the results at the click of a series of dropdown menus. Different scenarios can be tested simply by repeating the steps allowing for the sensitivity of the resultant temperature to the calibration methodology to be tested relatively quickly and easily. It would be nice if this sensitivity testing could be made (even) easier (e.g., selecting multiple tests prior to processing) but that would make it complex and I think the simplicity is ideal. All of this is wrapped up in a web based tool meaning that there is no code. Furthermore, to make everyone's job easier the authors have also incorporated an automatically generated (somewhat) detailed methodology text from the selected options. Making the tool useful for both experienced and new palaeoclimatologists. Such tools can help to contribute to open, transparent, and reproducible science. Therefore, I would recommend publication. However, I do have some recommendations regarding both the tool itself, the code, and the text.

The paper is split into three sections to describe the tool: a rationale (section 1); a description of the tool (section 2); and, a demonstration (section 3). The last section is where I have a bit of a problem as it reads like two technical notes stuck together (section 1 + section 2 vs. section 3). As in rather than the later section being an example for the tool described in this technical note this later section appears more focused on the error associated with the latitudinal method (i.e., see the juxtaposition between line 128-131 and lines 132-140). Reading more like a technical note on 'stop using the latitudinal method'. For a demonstration I would expect to read more about the rationale for the choices of the tool (line 94-98 as line 99- 104 is a reworded repeat of lines 67-73), what a user can expect, and what they can do rather than choices of data and data-comparison (line 111-115). I don't have a problem with the results of this section and its perfectly fine to keep in. I just feel like there are features of the tool and sections from the information section of the webpage that are not included for the sake of brevity and so this section could

be shortened/reworked. Whilst the instructions to authors for a 'technical note' do explicitly state that it shouldn't be a technical manual - so I understand the author's rationale - I do believe that pertinent information regarding the tool should be included in the 'scientific record' (in the paper or as supplementary materials). For instance, a summary of the various palaeotemperature equations, calibration limits, etc. (for an example see Figure 1 in the supplement to this review) as a table in this text would allow users to select the right palaeotemperature equation(s) rather than exploring this component of the tool 'at random'. Likewise, assumptions, caveats, and computational tricks that are in the code should be explained to aid transparency and avoid it being a blackbox. I would recommend (see major comments below) that the authors incorporate the comments that can be found in the code into the text, ensure that the method section includes all references including reformulations. As well as consider things like versioning, to ensure that the users can keep up with changes to the tool.

> We thank the reviewer for their thoughtful and highly constructive feedback. We have implemented the majority of their suggestions as-is, with specific comments below.

> With regard to the specific suggestion of a calibration table, we have added a table listing some of the most commonly used calibrations implemented by the model. We have not included all 62 calibrations from the tool in this table because this would require several pages, but we can do so if the reviewer/editor prefers. In any case, the methods used to implement the other calibrations are now explained in-text, with the exact algebraic form used listed in the source code on GitHub.

Major comments

Comments in code. The proxy.php script on Github has a whole bunch of comments that should be in the text, as a supplement, etc. For instance lines 1228, 1216, 1301-3, 1430,... etc. all include pertinent information as to what your tool is doing yet some aspects do not sappear in the associated methodology text that appears post processing. As such it makes your tool less replicable (especially compared with someone not using your tool) simply by removing some steps or introducing pertinent details or caveats that are not alluded to elsewhere. That has the knock-on effect that a user's methodology section will also be missing steps.

> Relevant comments in the code have now been incorporated into the manuscript text or methodology output, as appropriate. This entailed enough revision to Section 2 (55-134) that it should be considered substantially rewritten; see the manuscript for details.

Reformulations. Similarly to the above comment, I do think you need to incorporate 'reformulated by' into the methodology text. For instance, the Kim and O'Neil (1997) equation is the equation reformulated by Bemis et al (1998), there are several reformulated versions in the literature with not all being equivalent (see Figure 1 of this review) so it is important for the user to know which they are using.

> The tool now specifies the reformulation used in its methodology output.

References. Likewise, as there is no citation limit (only higher APC) I would cite in this article all of the calibrations, reformulations, etc. that your tool uses (rather than '.etc') as this would also officially recognise those author's contributions to your tool. Reiterating the point above I would also include all references for reformulations, compilations - e.g., Wilmes (line 1430 in proxy.php) here and in the methodology.

> This has now been done (see specific comments below).

Workflow diagram. A workflow diagram could be useful to visually explain lines 56-89.

This has been added as the new Figure 2.

Versioning. It would be prudent to incorporate a versioning system and/or last update. Versioning is important for replication, if there is a mistake, correction, etc then users can quickly make changes. I can imagine that once processed an author will be unlikely to keep recalibrating so versioning is useful as a hint for users that they might want to recalibrate their previous datasets. Will a webpage be included that lists the changes to the code (rather than assuming a user will explore the file in github to track changes)?

The tool now specifies a version number in its methodology output, with "Releases" on GitHub used to archive past versions of the tool.

Bibtex. Can you include a bibtex version of the citations for the methodology rather than/as well the nicely formatted references that appear post processing. You could either just include an additional webpage purely with the references in bibtex format or have a download bibtex button. I ask because whilst you have already done much of a user's work for them, if you want to ensure they cite the original papers then you should make the reference list importable to a reference manager. That way users would have absolutely no reason not to cite the original papers.

This has been added to the tool.

Reference code for calibration. Could you give the output a code that encapsulates the various options, e.g., something like c1t1i1s1, i.e., c1 the first option of the calibration menu (malevich et al), t1 the first option of the timescale (Gradstein et al), first option of the ice volume (Rohling et al), and the first option of the correction method (Gaskell et al). So the user can refer directly to which options were selected as there are a lot of dropdown menu steps. By setting up your own reference code you would also ensure 'interoperability'/ 'replication' between users . Then authors could refer to it as " This calibration was performed online using option's c1t1i1s1 (Gaskell et al)." or "comparison between c1t1i1s1 and c1t2i1s1 shows little.."

This is an interesting suggestion which we will consider for future versions. The difficulty with implementing it as suggested is that the input fields take a variety of forms of input (both categorical and numerical) and use categorical options whose number and order cannot necessarily be guaranteed to remain constant across versions. It may be possible to export a longer "parameterization string" which would serve a similar purpose, but this would require reworking some of the existing design. Nevertheless, we thank the reviewer for this thought-provoking suggestion, which we will consider how to adapt in the future.

Geographically nearest. You average the nearest lat/lon as specified by the user, can we not get a range/stdev for this as well? I also assume that this is uses 'nanmean' to take into account missing values (e.g., land) so there will be variation in N depending on if the site +-lat/lon is open ocean versus close to the coast. Can you inform the user of this? Likewise, as mentioned in the minor comments below, latitude doesn't change distance but longitude does so using the same longitude plus/minus for different sites won't (always) be the same area averaged. Again it would be important the user realizes this.

These suggestions have been implemented in the tool, and an option to select d18Osw points by great circle distance rather than latitude/longitude has been added.

Minor comments

Line 6: could modify to: "However, interpretation of such data is complicated by the necessity of knowing the d18Osw of the source seawater from which CaCO3 is precipitated."

This wording change has been adopted (line 6).

Line 20: this is a linear calibration but many of the calibrations are quadratic. Would it thus not be more prudent to change this to:

"... empirical calibration in either a quadratic (eq 1) or linear (eq 2) form:

T = a - b*(d18Oc -d18Osw) - c*(d18Oc -d18Osw)^2 (1)

T = a - b(d18Oc -d18Osw) (2)"

We have added an example of a quadratic calibration (McCrea et al. 1950 as reformulated by Bemis et al. 1998; line 21)

Line 25: 'function of sea level' its a function of ice volume

"Ice volume" is now used instead of "sea level". (line 25)

Line 29: 'less reliable' I would be less diplomatic 'next to impossible compared with'

We are more inclined to be diplomatic, as a number of recent papers have shown good success at using d18O for SST measurements (Gaskell et al. 2022 PNAS; de Vleeschouwer et al. 2019 Clim. Past; etc.). However, this remains a topic of debate in the proxy community, and we hope that by improving the accessibility of the best available methods, our tool will help to further clarify the issue.

Line 36-41: refer to changes in E:P

We have added "due to alterations in the local ratio of evaporation to precipitation" to the explanation in this section. (line 37)

Line 56: could change to: "Here a new online tool for performing d18Oc-temperature conversion is presented that automates a range…"

This wording change has been adopted. (line 53)

Line 58: state format/extension of dataset, "After manually entering or uploading a .csv datasheet…" or "After manually entering or uploading a datasheet of d18Oc in .csv format…"

This has been changed to "After manually entering or uploading a datasheet of $\delta^{18}O_c$ measurements in .csv format…" (line 62)

Line 60: remove 'etc' and cite all the calibrations paper and reformulation so as to officially recognise those author's contributions to your tool.

The full list of citations has been added in-text. Full citations have also been added to line #####, for the methods of calculating global seawater d18O. (lines 63-67; 83-84)

Line 67-71: this is repeated at line's 99-104. I would move the text from 99-104 here and not repeat it in section 3.

(n/a since section 3 has been omitted)

Line 86-89: state somewhere here that the tool also informs the user if the temperature exceeds the calibration limits.

This has been changed to "Any rows with potential errors (e.g., paleocoordinates which do not yield a valid $\delta^{18}O_w$ estimate or temperatures which exceed the data range of the calibration) are flagged with a warning." (line 126)

Line 92/93: Include that you are focusing on the Late Paleocene, Eocene (prior to line 98), and PETM (prior to line 105) for this example from the onset.

(n/a since section 3 has been omitted)

Line 104: (also line 73-74) latitude is constant but longitude changes distance, how is this accounted for?

This method behaves as described in Gaskell et al. (2022) PNAS, which our implementation replicates. The reviewer is correct that the dimensions of the patch do not remain constant across all latitudes; the tool now includes an option to calculate patches by great-circle distance, as is currently done for the nearest-point option.

Line 123: change to 'that are significantly closer to the'

(n/a since section 3 has been omitted)

Line 128: 'explore the sensitivity' point out to the reader that they have to re-run the tool to explore the sensitivity as there is no method to automatically re-run for different scenarios.

We have added the text "by successively trying different options in the interface" to this line. (line 136)

Code/data

Data availability statement of the DeepMip 0.1 proxy database

(n/a since section 3 has been omitted)

Clean/tidy up the code, i.e., remove FIXME comments

This has been done.

References

The same references are used for the tool so the following should be checked: Gradstein et al. (2020) either publisher city (San Diego) or country (The Netherlands) is incorrect. Sharp (2017) has 2nd edition twice; Cramer et al, Gaskell et al, Malevich et al, Zhou et al all need superscript for the d18O; Ogg et al needs publisher city

These have been corrected.

Figures

Figure 1: the lower end of the temperature scale and land are a bit hard to distinguish. Can you not replace the land with a different color? (e.g., to gray)

This has been done.

Figure 2: is not cited in the text. I would also include a symbol/shading/feature to allude to 'no-data for this time period' ; it is not apparent to the reader when quickly glancing. Also include that missing data means lack of data in the caption.

(n/a since section 3 has been omitted)

Figure 2: why not plot as a difference? You could choose one as a 'reference' compute the difference from that.

(n/a since section 3 has been omitted)

Figure 3: d18Oc-based temperature is plotted against combined Mg/Ca and Tex86 is there a difference between the d18Oc-based temperature vs. Mg/Ca or vs. Tex86? For me age in the left plot is not needed so you could make the orange and blue Mg/Ca and Tex86 rather than LP and PETM.

(n/a since section 3 has been omitted)

**REVIEWER COMMENT #2**

Gaskell and Hull present a useful online tool that allows to "play" with different d18O temperature conversions and correction procedures. I think this is a convenient way to evaluate the impact of these different procedures on oxygen isotope datasets. In addition to the tool they also aim to demonstrate that common approaches lead to temperature underestimations in the the North Atlantic and overestimates in the Southern Ocean during hothouse climates.

We thank the reviewer for their positive and constructive feedback. Specific responses and edits related to their comments are listed below.

The tool works smoothly and generates a nicely organised output. It would be great if several options could be selected at the same time to allow quick direct comparison of the effect of different options. While this would be a nice addition I would not say it is a must have.

This is a common request, but probably something we will leave for a future version as it involves several technical challenges (e.g., maximum PHP runtime defined by the server) and would substantially increase the complexity of the interface for a relatively small gain in usability. At present, results can be compared manually by the user.

In the conclusion I would suggest to not to focus on the latitude effect alone but instead summarise the various effects that different correction steps/assumptions on oxygen isotope derived temperatures have in general. "Ground-truthing" any d18O derived temperatures with either Mg/Ca and or TEX86 is not fully appropriate as both of these proxies have their own uncertainties, require assumptions and have complications. If done it would require a thorough discussion of these as well. I think the comments on the North Atlantic specifically could be omitted and would be better placed in a separate article and not necessarily in the presentation of this tool. I would therefore also suggest to exclude the part on the North Atlantic and Southern Ocean (line 12-16) from the abstract.

In response to feedback from both reviewers, we have omitted the majority of the former Section 3 from this revision. Our comments regarding the North Atlantic are now restricted to the discussion of Figure 1 in the Introduction (generally lines 31-42).

Line 25: Sharp 2017 is not an appropriate reference here.

This has been replaced with a reference to Rohling et al. (2013), which covers these statements more comprehensively than they are covered by Sharp (2017). (Line 26)

Line 27: The introduction gives a nice overview, however I am missing references to highly relevant new insights from clumped isotope thermometry that can be used in addition to classic d18O measurements.

E.g. Clumped isotope estimates challenge the simple conversion of deep-sea oxygen isotopes to temperature (see discussions in Agterhuis et al. 2022 and Meckler et al., 2022).

> We thank the reviewer for suggesting these references, particularly Meckler et al. (2022), whose results we have added as a new d18Osw record to our tool. Since the recent clumped-isotope work primarily challenges prior records of d18Osw (by changing what bottom-water temperature is subtracted from the benthic d18O stack to obtain d18Osw), we have added a brief discussion of this to section 2.2: "Which global $\delta^{18}O_w$ record is most realistic remains a contentious topic in the literature, with sea-level and Mg/Ca-based records (e.g., Cramer et al., 2011; Rohling et al., 2021 predicting up to ~1‰ lower $\delta^{18}O_w$ for much of the Cenozoic than records based on clumped isotope paleothermometry (Meckler et al., 2022; see also Agterhuis et al., 2022). We provide both classes of record here for comparison by the user." (lines 87-90)

Line 31-41: What about uncertainties in seasonally variable d18Ow (associated with often unknown/uncertain calcification season of the planktic foraminifera)

> Seasonal variation in surface d18Osw has generally been omitted from most published temperature-conversion methods. This is largely because seasonal variations in d18Osw are typically small compared with seasonal variations in temperature, which in turn are convolved with seasonal variations in foram abundances, making seasonality of d18Osw a relatively minor component of the overall uncertainty. Core-top calibrations such as bayfox are constructed by comparing annually-averaged forams (such as core tops) to annually-averaged d18Osw (such as that estimated by LeGrande & Schmidt 2006, Tierney et al. 2020, etc.), which avoids the problem altogether by implicitly baking seasonal variations in surface d18Osw into the calibration equation and its corresponding uncertainty bounds.

> A better treatment of this issue in the literature is needed, but probably beyond the scope of this Report, which focuses on implementing existing methods rather than developing new ones. However, we have added a brief mention of this to section 2.3: "Our tool does not currently implement any automated consideration of seasonal variation in local $\delta^{18}O_w$, as this is generally treated as negligible by standard methodologies or implicitly baked into the calibration by calibrating against mean annual temperatures and $\delta^{18}O_w$ values (e.g., Malevich et al., 2019).". (lines 111-114)

line 69: specify where the local d18Osw data is derived from (Tierney 2020 model output?)

> Yes; this has been reworded to "(model output from Tierney et al., 2020)". (line 101)

line 71: for the bottom water temperature it would be again to include the results of Meckler et al. here as well

> We have added the Meckler et al. (2022) clumped-isotope temperature curve (and associated raw and pH-corrected d18Osw curves) to the tool, using the Monte Carlo LOESS method described in the paper for age interpolation. This is now described on lines 91-93.

line 82: in the tool, it would be nice to see immediately what the different options for the 's' values are

> These have been added to the menu options.

line 97: Meckler et al. derive different d18O sw values using clumped isotopes, would be useful to include these values also in the tool but also use it in this example

(this has now been added to the tool; see above)

line 97-98: for the purpose of this article demonstrating the tool it would be nice to show in the example what the potential effect is of carbonate ion effect on final temperatures

(n/a since section 3 has been omitted)

line 132-140: In the conclusion I would suggest to avoid focussing here on the latitude but instead summarise and highlight the large variability that can derive from using different assumptions in oxygen isotope thermometry

(n/a since section 3 has been omitted)

---

## Referee Report (RR1)

**Technical note: A new online tool for δ18O-temperature conversions. Review: B. Metcalfe**

The authors have made the recommended changes or sufficiently explained why the comments of the previous review either no longer matter (e.g., removing section 3) or inconsequential to the current paper. I recommend publication as is, with only the following textual modifications (these could also be done at the proof reading stage):

Line 119: "of the carbonate, δ18Oc' is" the second comma is wrong?

Line 127: maybe clarify here what flag, i.e., just add (NaN) or (-99)?

Line 347: the bracket before ( doesn't have its partner, and shouldn't 2012 be (2012)?

Code availability section: I would add the licence type and DOI to this section, i.e., "Source code (Javascript and PHP) is openly available under a – *name of licence* - license from the project's GitHub repository at … (DOI: - *doi*)"  with the persistent identifier DOI obtainable via [1].

**Licencing**

I would add a 'licence.txt' file to your GitHub (LICENSE.txt; LICENSE.md or LICENSE.rst) in the root of the repository not just adding the licence in the README or in proxy.php, as per [2] it is considered a "… best practice, we encourage you to include the license file with your project".

**Note**

Inconsequential to the paper but in the Author response to the other reviewer (pg 7) you state that: "*Core-top calibrations such as bayfox are constructed by comparing annually-averaged forams (such as core tops) to annually-averaged d18Osw (such as that estimated by LeGrande & Schmidt 2006, Tierney et al. 2020, etc.), which avoids the problem altogether by implicitly baking seasonal variations in surface d18Osw into the calibration equation and its corresponding uncertainty bounds.*". It is an assumption that this implicitly bakes it in, but core tops are not annually averaged per se. They can be seasonally skewed based upon ecological preferences of individual foram species and this will vary in terms of where the sample is taken with respect to a species ecological space/biography. If the species ecological window overlaps with the core tops environmental range then it can be considered to represent an annual signal (although even then that assumes there is not a flux, sampling, or picking bias), but in most regions it will be biased toward one season or another (see Mix (1987); Mulitza et al., (1997); or Pracht et al (2019)). Worse there is also depth habitat to consider, which may vary also regionally. So, I would respectfully disagree that core top calibrations 'implicitly bake in seasonal variations' by using annual averaging unless they implicitly take into account season and depth which few do.

**References**

[1]     https://docs.github.com/en/repositories/archiving-a-github-repository/referencing-and-citing-content

[2]                 https://docs.github.com/en/repositories/managing-your-repositorys-settings-and-features/customizing-your-repository/licensing-a-repository

---

## Author Response (AR2)

The wording changes suggested by the referee below have been adopted. We have also updated Table 1 to include all relevant equations, as requested by the editor.

**Reviewer 1 comments**

Technical note: A new online tool for δ18O-temperature conversions. Review: B. Metcalfe

The authors have made the recommended changes or sufficiently explained why the comments of the previous review either no longer mater (e.g., removing section 3) or inconsequential to the current paper. I recommend publication as is, with only the following textual modifications (these could also be done at the proof reading stage):

Line 119: "of the carbonate, δ18Oc' is" the second comma is wrong?
Line 127: maybe clarify here what flag, i.e., just add (NaN) or (-99)?
Line 347: the bracket before ( doesn't have its partner, and shouldn't 2012 be (2012)?
Code availability section: I would add the licence type and DOI to this section, i.e., "Source code (Javascript and PHP) is openly available under a – name of licence - license from the project's GitHub repository at ... (DOI: - doi)" with the persistent identifier DOI obtainable via [1].

Licencing: I would add a 'licence.txt' file to your GitHub (LICENSE.txt; LICENSE.md or LICENSE.rst) in the root of the repository not just adding the licence in the README or in proxy.php, as per [2] it is considered a "... best practice, we encourage you to include the license file with your project".

Note

Inconsequential to the paper but in the Author response to the other reviewer (pg 7) you state that: "Core-top calibrations such as bayfox are constructed by comparing annually-averaged forams (such as core tops) to annually-averaged d18Osw (such as that estimated by LeGrande & Schmidt 2006, Tierney et al. 2020, etc.), which avoids the problem altogether by implicitly baking seasonal variations in surface d18Osw into the calibration equation and its corresponding uncertainty bounds.". It is an assumption that this implicitly bakes it in, but core tops are not annually averaged per se. They can be seasonally skewed based upon ecological preferences of individual foram species and this will vary in terms of where the sample is taken with respect to a species ecological space/biography. If the species ecological window overlaps with the core tops environmental range then it can be considered to represent an annual signal (although even then that assumes there is not a flux, sampling, or picking bias), but in most regions it will be biased toward one season or another (see Mix (1987); Mulitza et al., (1997); or Pracht et al (2019)). Worse there is also depth habitat to consider, which may vary also regionally. So, I would respectfully disagree that core top calibrations 'implicitly bake in seasonal variations' by using annual averaging unless they implicitly take into account season and depth which few do.

References
[1] htps://docs.github.com/en/repositories/archiving-a-github-repository/referencing-and-citing-content
[2] htps://docs.github.com/en/repositories/managing-your-repositorys-settings-and-features/customizing-your-repository/licensing-a-repository